# One-Pot Fabrication of 2D/2D CdIn$_2$S$_4$/In$_2$S$_3$ Heterojunction for Boosting Photocatalytic Cr(VI) Reduction

**Jiawei Hu, Jiaxin Wu, Siyuan Zhang, Wenxuan Chen, Wen Xiao, Haijun Hou, Xiaowang Lu \*, Chao Liu \*** 
**and Qinfang Zhang**

School of Materials Science and Engineering, Yancheng Institute of Technology, Yancheng 224051, China; qfangzhang@gmail.com (Q.Z.)
\* Correspondence: luxiaowang@ycit.edu.cn (X.L.); cliu@ycit.edu.cn (C.L.)

**Abstract:** The development of efficient heterojunction photocatalysts with a facilitated charge carrier separation rate and improved light-harvesting capacity is still a challenging issue for effectively solving environmental pollution. Herein, a one-step refluxing process was employed to construct 2D/2D CdIn$_2$S$_4$/In$_2$S$_3$ (CISI) heterojunction photocatalysts with an intimate interface between these two components. The crystal structure, morphology, light-harvesting capacity, and Cr(VI) photoreduction performance were systematically investigated and discussed in detail. The tight interface formed between CdIn$_2$S$_4$ (CIS) and In$_2$S$_3$ (IS) could effectively facilitate the charge carrier separation and transfer. Thus, the resulting CISI composites exhibited a high efficiency for Cr(VI) photoreduction under visible light, with the optimal sample of 0.5 CISI. The charge transfer kinetics were deeply investigated by multiple techniques. Based on the characterization results, a possible mechanism for Cr(VI) photoreduction was proposed.

**Keywords:** CdIn$_2$S$_4$; In$_2$S$_3$; photocatalysis; heterojunction; Cr(VI) photoreduction

## 1. Introduction

Over the past decades, the presence of the toxic heavy metal hexavalent chromium (Cr(VI)) in liquid industrial waste has had a significantly negative impact on the ecological system and human health on account of its acute toxicity, good solubility, and high carcinogenicity [1,2]. Conversely, trivalent chromium (Cr(III)), with a lower toxicity, is considered to be the essential microelement for the metabolism of various organisms. Furthermore, Cr(III) favors the appearance of sedimentation under alkaline and neutral conditions. Thus, the reduction of Cr(VI) to (Cr(III) is regarded as an effective pathway for the removal of Cr pollution. Until now, semiconductor-based photocatalytic technology has been extensively applied in the elimination of pollutants due to its low cost, environmental friendliness, and extraordinary efficacy [3–7]. For the photoreduction of Cr(VI) to Cr(III), constructing high-efficiency photocatalysts with a high charge carrier separation rate and a light harvesting capacity is imperative to meet the demand of industrial development.

Recently, diverse semiconductor-based photocatalysts, such as metal sulfides, TiO$_2$ [8,9], layered double hydroxides (LDHs) [10], WO$_3$ [11], and BiPO$_4$ [12], have been developed for Cr(VI) photoreduction. Among these photocatalysts, metal sulfides have attracted ever-increasing attention in various photocatalytic fields on account of their simple synthesis process, low cost, visible-light harvesting capacity, and unique photoelectric properties [13,14]. In particular, the ternary semiconductor CdIn$_2$S$_4$ (CIS) has been extensively investigated due to its easy accessibility, structural stability, and suitable energy band structure [15,16]. Unfortunately, like most single photocatalysts, the rapid charge carrier recombination rate of CIS restrains its potential application in achieving the desirable photocatalytic efficiency [17].

Generally, photocatalytic efficiency is profoundly dependent on the charge carrier separation rate. To meet the practical requirements, some modified strategies have been

used to suppress the charge recombination in order to boost the photocatalytic efficiency of single photocatalysts. These modification methods mainly include heterojunction formation [18–20], morphology design [21–23], elemental doping [24,25], cocatalyst deposition [26,27], and defect engineering [28,29]. CIS-based heterostructured photocatalysts, in particular, could significantly accelerate the charge separation and thus improve the photocatalytic activity of CIS, including $CIS/ZnIn_2S_4$ for water splitting and $CO_2$ reduction [28], carbon quantum dots/CIS for photodegradation [30], $TiO_2/CIS$ for photocatalytic hydrogen evolution [31], etc. However, it is still a significant challenge to construct an effective heterojunction with efficient charge carrier separation and highly photocatalytic activity.

For this purpose, herein, a one-step fluxing method was developed to successfully construct 2D/2D $CdIn_2S_4/In_2S_3$ (CISI) heterostructured photocatalysts for visible light-driven Cr(VI) photoreduction. 2D CIS and 2D IS were chosen as primary components to fabricate a 2D/2D CISI heterojunction for Cr(VI) photoreduction based on the following reasons: (1) The suitable energy band structures of CIS and IS show visible light responses and sufficiently strong redox potentials. (2) The one-step synthesis method can be regarded as an advantageous approach to prepare heterojunction photocatalysts compared to other synthesis preparation methods. (3) The constructed 2D/2D heterojunction created via the one-step synthesis method has the maximum amount of contact interfaces between two components, resulting in the efficient separation of photogenerated charge carriers. The resulting 2D/2D CISI heterostructured photocatalysts show a high efficiency for Cr(VI) photoreduction under visible light. A possible mechanism for Cr(VI) photoreduction is also speculated.

## 2. Results

### 2.1. Synthesis of $In_2S_3$ (IS)

To prepare $In_2S_3$ (IS), 4 mmol of $InCl_3 \cdot 4H_2O$ and 6 mmol of thioacetamide (TAA) were dissolved in 100 mL of deionized water. After stirring for 30 min, the mixed solution was transferred to a three-necked flask (250 mL) and then heated at 90 °C for 5 h in an oil bath. After cooling down to room temperature, the obtained product was filtrated and washed with deionized water several times and dried at 60 °C overnight.

### 2.2. Synthesis of $CdIn_2S_4$ (CIS)

In a typical process, 1.2 mmol of $Cd(CH_3COO)_2 \cdot 2H_2O$, 2.4 mmol of $InCl_3 \cdot 4H_2O$, and 6.4 mmol of thioacetamide (TAA) were added into 200 mL of deionized water. After constant stirring for 30 min, the uniform solution was transferred to a 250 mL three-necked flask for heating at 100 °C for 12 h in an oil bath. Afterwards, the obtained precipitate was centrifuged, washed with water several times, and dried in an oven at 60 °C overnight.

### 2.3. Synthesis of $CdIn_2S_4/In_2S_3$ (CISI) Composites

The $CdIn_2S_4/In_2S_3$ (x-CISI) composites (x = 0.25, 0.5, and 0.75) were synthesized by the similar method to that of CIS, where x represented the molar ratio of $In_2S_3$ in CISI composites. An amount of $2 - 2x$ mmol of $Cd(CH_3COO)_2 \cdot 2H_2O$, 4 mmol of $InCl_3 \cdot 4H_2O$, and $12 - 3x$ mmol thioacetamide (TAA) were dissolved in 150 mL deionized water under constant stirring for 30 min. Subsequently, the mixed solution was transferred to a 250 mL three-necked flask and then heated at 100 °C for 12 h in an oil bath. After cooling to an ambient temperature, the precipitates were collected by centrifugation, washed with deionized water several times, and dried in an oven at 60 °C overnight.

### 2.4. Characterization

To observe morphology, scanning electron microscopy (SEM) images were recorded in Nova Nano SEM 450 (Hillsboro, OR, USA). Additionally, transmission electron microscopy (TEM, JEOL JEM-2100F, Tokyo, Japan) with an accelerating voltage of 200 kV was performed to confirm microstructure and elemental distribution. The powder X-ray diffractometer

(XRD, PANalytical B.V., Almelo, Netherlands) with Cu K$\alpha$ radiation ($\lambda$ = 0.15418 nm) was used to verify the crystal structure of samples. To determine light absorption capacity, UV–vis diffuse reflectance spectra (DRS) were measured on a UV–vis spectrophotometer (Shimadzu UV-3600plus, Kyoto, Japan) using $BaSO_4$ as a reference. X-ray photoelectron spectrometer (Thermo Scientific, Escalab 250 Xi, Waltham, MA, USA) with Al K$\alpha$ X-ray source (h$\nu$ = 1486.6 eV) was employed to confirm the atomic states of photocatalysts using a hemispherical electron analyzer (pass energy of 20 eV). The adventitious C 1 s peak at 284.8 eV acted as the corrected binding energy. Photoluminescence (PL) spectra were recorded on luminescence spectrometer (F-4600, Tokyo, Japan) using the excitation wavelength of 400 nm. Time-resolved photoluminescence (TR-PL) decay spectra were obtained on a fluorescence spectrometer (FLS1000, Edinburgh, UK) using the excitation wavelength of 330 nm.

*2.5. Photo-Electrochemical Measurements*

The photo-electrochemical data were recorded in an electrochemical workstation (CHI-660E, Shanghai Chenhua, China) with a three-electrode cell. The platinum (Pt) wire and Ag/AgCl were used as the counter electrode and the reference electrode, respectively. The working electrode was gained by the following procedure: An amount of 1.0 mg photocatalyst was mixed with 1 mL Nafion (0.5%). After ultrasonic treatment, the 40 $\mu$L uniform slurry was sprayed onto fluorine-doped tin oxide (FTO) glass (1 $\times$ 1 cm$^2$) and then naturally dried. Photocurrent was measured under a 300 W Xenon arc lamp ($\lambda$ > 420 nm) in $Na_2SO_4$ aqueous solution (0.2 M, pH = 7.0). Linear sweep voltammetry (LSV) and Mott–Schottky (M-S) measurements were performed in $Na_2SO_4$ aqueous solution (0.2 M, pH = 7.0). M-S measurements were performed at a frequency of 1000 Hz. Electrochemical impedance spectra (EIS) were conducted in a mixed solution of KCl (1 mol/L), $K_3[Fe(CN)_6]$ (0.02 mol/L), and $K_4[Fe(CN)_6]$ (0.02 mol/L) at the open circuit potential.

*2.6. Photocatalytic Cr(VI) Reduction*

The photocatalytic activity of samples were evaluated by Cr(VI) reduction under a 300 W Xenon arc lamp ($\lambda$ > 420 nm). For a typical process, 10.0 mg photocatalyst and 1 mL citric acid (100 mg/L) were added into Cr(VI) solution (100 mL, 20 mg/L), and then vigorously stirred for 30 min in the dark to reach adsorption–desorption balance. After visible light irradiation, the above suspension was sampled with an interval of 15 min. According to the 1,5-diphenylcarbazide method, the absorbance of Cr(VI) solution was determined on a Shimadzu UV-2250 spectrophotometer at the wavelength of approximately 540 nm. The efficiency for Cr(VI) photoreduction was confirmed by the value of $C_t/C_0$, where $C_t$ and $C_0$ are the concentration at time t and the initial concentration, respectively.

**3. Discussion**

The morphologies of the as-prepared samples were characterized via scanning electron microscopy (SEM) (Figure 1). Clearly, the irregular particles can be observed over CIS and IS due to the agglomeration of their nanosheets (Figure 1a,b), which is confirmed by the below TEM images. Furthermore, the particular morphologies of the CISI composites are similar to that of CIS and IS (Figure 1c–e), indicating the maintenance of the overall structure for the CISI composites.

In the transmission electron microscopy (TEM) image in Figure 2a, CIS nanosheets are visible with some overlayers. Additionally, the sample of IS has the nanosheet-contained structure with the agglomeration phenomenon (Figure 2b). After the combination of CIS with IS, the similar nanosheet-contained structure also achieved over 0.5 CISI compared to IS (Figure 2c). As displayed in Figure 2d,e, the lattice distances of CIS and IS are estimated to be 0.38 nm and 0.62 nm, corresponding to the (220) plane of CIS and the (111) plane of IS, respectively, based on the crystallographic symmetry [32,33]. These two lattice fringes with the spacing value of 0.38 nm and 0.62 nm appear in 0.5 CISI, indicating the successful combination between CIS and IS (Figure 2f). Meanwhile, the

formation of an intimate interface between CIS and IS (red line) shows the successful construction of an effective heterojunction, which benefits the efficient separation of the photogenerated charge carriers [34,35]. The scanning transmission electron microscopy high-angle annular dark-field (STEM-HAADF) (Figure 2g) and the corresponding elemental distribution images (Figure 2h–j) of 0.5 CISI show the uniform distribution of Cd, In, and S, further confirming a close interface between CIS and IS.

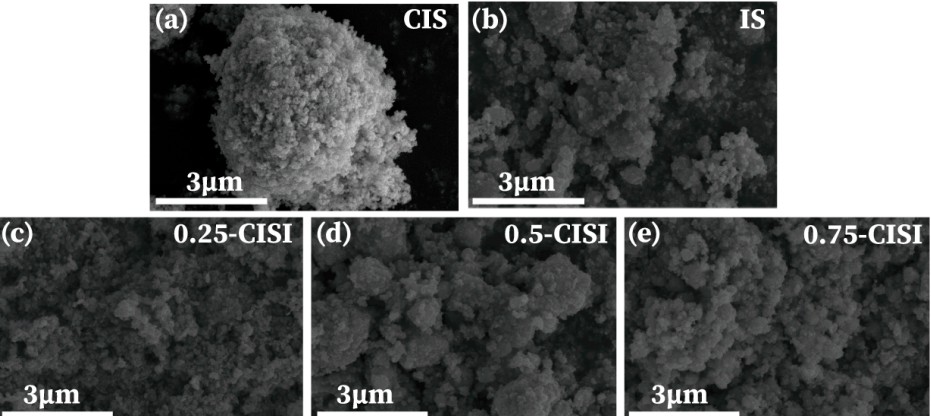

**Figure 1.** This SEM images of (**a**) CIS, (**b**) IS, (**c**) 0.25 CISI, (**d**) 0.5 CISI, and (**e**) 0.75 CISI.

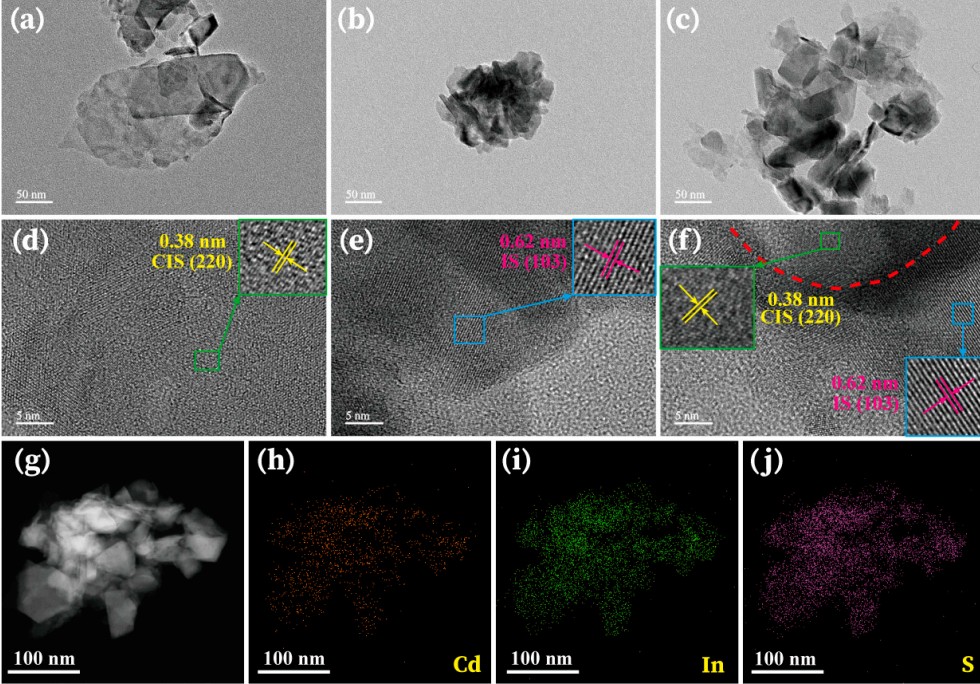

**Figure 2.** TEM images of (**a**) CIS, (**b**) IS, and (**c**) 0.5 CISI. HRTEM images of (**d**) CIS, (**e**) IS, and (**f**) 0.5 CISI. (**g**) STEM-HAADF image and (**h**–**j**) the distribution of Cd/In/S on 0.5 CISI.

The crystal structure of the as-prepared samples was monitored by XRD measurements. As shown in Figure 3a, the characteristic peaks of CIS and IS are well matched with their respective standard crystal structures of PDF#27-0060 and PDF#25-0390. Notably, the characteristic peaks of CIS are similar to those of IS except for the differences at 27.3 and 33.0°. After coupling CIS with IS, the characteristic peaks of 0.25 CISI, 0.5 CISI, and 0.75 CISI are almost unchanged related to CIS and IS originally. For the enlarged XRD patterns (Figure 3b), the characteristic peaks of 0.25 CISI and 0.5 CISI are derived from CIS, while the characteristic signals of 0.75 CISI are stemmed from IS. Thus, it can be

reasonably concluded that the CISI composites have been successfully synthesized with the components of CIS and IS.

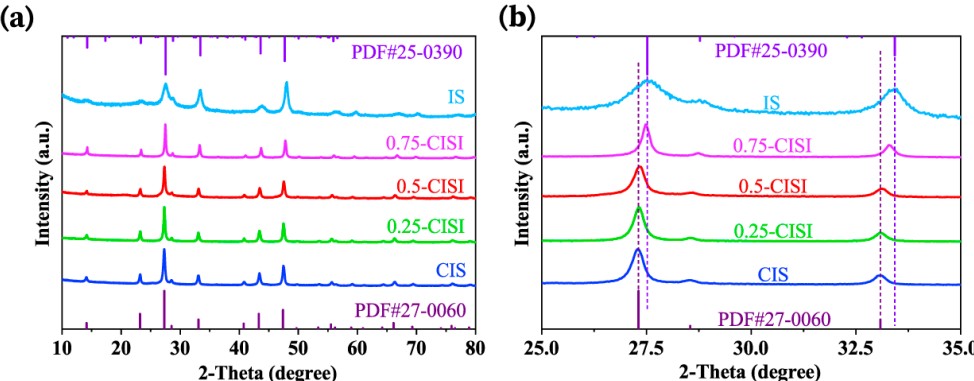

**Figure 3.** (**a**) XRD patterns and (**b**) enlarged XRD patterns of as-prepared samples.

The optical properties of CIS, IS, and CISI composites were investigated (Figure 4a). The absorption band edges of IS and CIS are at 561 and 616 nm, respectively, showing good visible light harvesting capacity. After coupling CIS with IS, the light absorption capacities of the CISI composites are improved compared to the single IS. Meanwhile, based on the Kubelka–Munk function [36,37], the bandgap values of CIS and IS are calculated to be 2.47 and 2.23 eV, respectively.

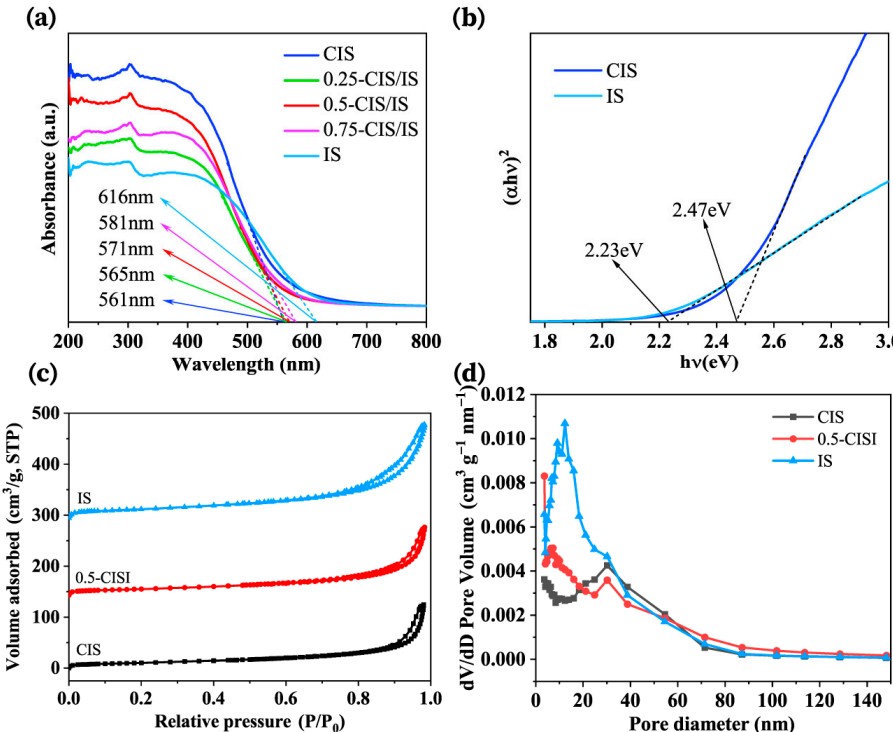

**Figure 4.** (**a**) UV–vis diffuse reflectance spectra (DRS) of samples. (**b**) Band gap energies of CIS and IS. (**c**) The $N_2$ absorption–desorption isotherms of CIS, IS, and 0.5 CISI. (**d**) Pore-size distribution plots of CIS, IS, and 0.5 CISI.

As depicted in Figure 4c, all the $N_2$ adsorption–desorption isotherms can be classified as type IV isotherms with an H3 hysteresis loop at $P/P_0$ between 0.4 and 1.0, suggesting the mesoporous materials for CIS, IS, and 0.5 CISI. The pore-size distribution curves also confirm the above mesoporous materials (Figure 4d) [38]. The Brunauer-Emmett-Teller (BET) surface area values ($S_{BET}$) of CIS, IS, and 0.5 CISI are measured to be 37.3, 61.9, and

44.1 m$^2$/g, respectively. Specifically, after combining IS nanoflakes with CIS nanosheets, the S$_{BET}$ of 0.5 CISI is reduced due to their nanosheets stacking between CIS and IS [39]. Generally, a higher surface-to-volume ratio of the photocatalyst can offer more active sites. Thus, the relatively high BET surface areas of all samples benefit Cr(VI) photoreduction.

The X-ray photoelectron spectroscopy (XPS) technique was employed to determine the elemental composition and atomic states of CIS, IS, and 0.5 CISI. As shown in Figure 5a, the XPS survey spectra of CIS and IS demonstrated that they mainly contain elements of Cd, In, and S, while the sample of IS has no signals of Cd. As for the XPS of the Cd 3d spectrum of CIS (Figure 5b), two strong peaks appear at 411.8 and 405.1 eV, corresponding to Cd 3d3/2 and Cd 3d5/2, respectively [29]. However, the peak positions in 0.5 CISI are positively shifted to 412.2 and 405.3 eV. This indicates that the elemental valences of Cd in CIS and 0.5 CISI are +2 [40]. The XPS In 3d spectrum of CIS shows two peaks centered at 452.3 eV and 444.7 eV (Figure 5c), which are assigned to In 3d3/2 and In 3d5/2, respectively [41]. These two characteristic peaks also appear in pure IS. After coupling CIS with IS, there is an obviously positive shift for XPS iIn the 3d spectrum of 0.5 CISI compared to CIS and IS. As shown in Figure 5d, the S 2p XPS spectra of CIS and IS possess similar signals at ~162.6 and ~161.4 eV, which belong to S 2p1/2 and S 2p3/2, respectively [42]. Clearly, these two peaks of XPS S 2p in 0.5 CISI also exhibit a positive shift related to CIS and IS. Combined with the XPS results of Cd 3d and In 3d, we reasonably conclude that there is a strong interaction existing in the CIS/IS heterojunction, which is conducive to the efficient charge carrier transfer [43].

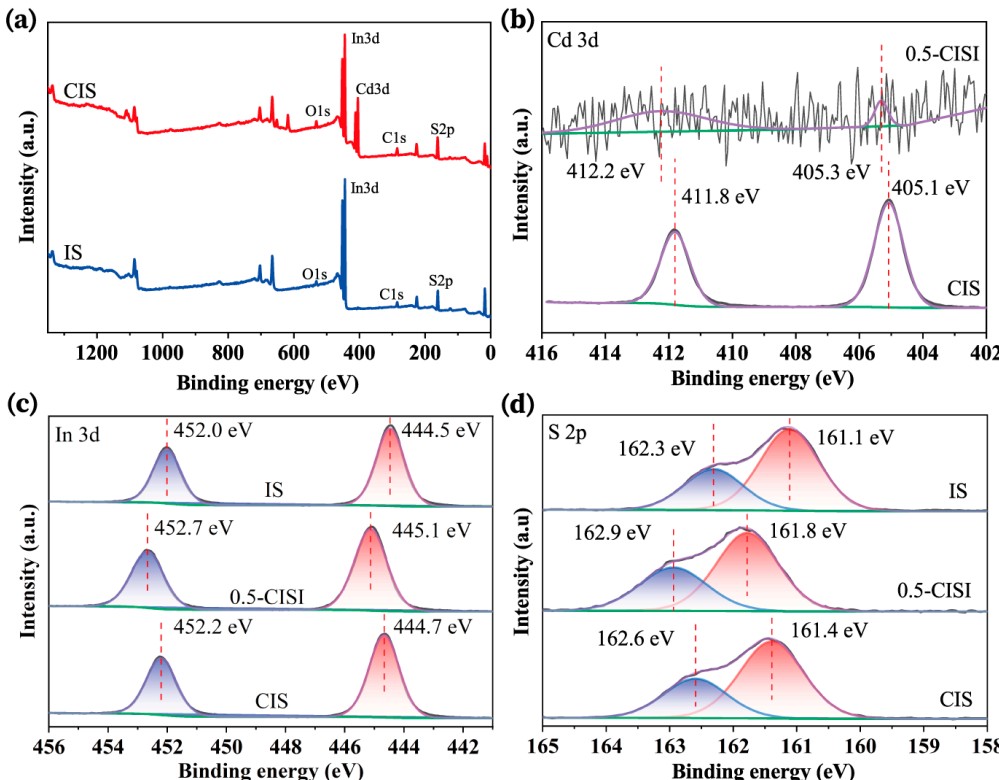

**Figure 5.** High-resolution XPS spectra of CIS, IS, and 0.5 CISI: (**a**) survey, (**b**) Cd 3d, (**c**) In 3d, and (**d**) S 2p.

The steady-state photoluminescence (PL) spectra of the as-prepared samples were employed to investigate the charge separation behavior (Figure 6a) [44]. The PL peak intensities of 0.25 CISI, 0.5 CISI and 0.75 CISI are weaker than those of CIS and IS, demonstrating that the heterojunction formation between CIS and IS can effectively promote the charge separation. Additionally, the highest PL intensity being that of the 0.5 CISI sample suggests that the proper mass ratio between CIS and IS can achieve the effective heterojunction for

efficient charge carrier separation, resulting in the improved photocatalytic activity for Cr(VI) reduction.

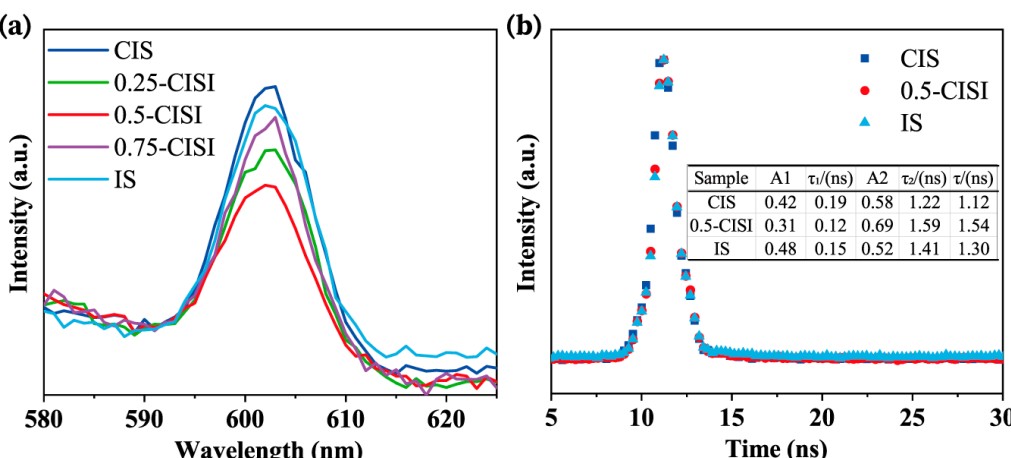

**Figure 6.** (**a**) PL spectra with the excitation wavelength of 400 nm and (**b**) TR-PL decay spectra of as-prepared samples.

The time-resolved PL (TR-PL) decay curves in Figure 6b also support the above results. The fluorescence lifetime is closely related to the charge carrier separation rate [45,46]. Generally, a longer fluorescence lifetime means a higher charge separation efficiency. Based on our previous work [37], the average lifetimes of CIS, IS, and 0.5 CISI are calculated to be 1.12, 1.30, and 1.54 ns, respectively. Obviously, the average lifetime of 0.5 CISI is higher than that of CIS and IS, reflecting that the heterojunction formation between CIS and IS can prolong the lifetime of photogenerated charge carriers and thus boost activity for Cr(VI) photoreduction.

The electrochemical impedance spectra (EIS) and the transient photocurrent responses (TPRs) were performed to reveal the charge separation and migration behavior of the samples [47–49]. As for the EIS measurements, the smaller arc radius is closely related to the lower interfacial resistance, resulting in a higher charge migration efficiency in the Nyquist plots. As shown in Figure 7a, the arc radii of 0.25 CISI, 0.5 CISI, and 0.75 CISI are smaller than those of CIS and IS, demonstrating that the formation of the heterojunction can effectively reduce the interfacial resistance and thus promote the charge carrier separation. The optimal sample, 0.5 CISI, has the smallest arc radius, which means the highest migration rate of photogenerated charge carriers in all CISI composites.

The TPR spectra of samples are displayed in Figure 7b for a total of nine on–off photoperiod tests. Pure CIS and IS show relatively low photocurrent intensities due to their rapid recombination rate of photogenerated charge carriers, leading to the low availability of photogenerated electrons [50]. When CIS was coupled with IS to construct heterostructured photocatalysts, all CISI composites exhibited the increased TPR intensity, suggesting that the recombination rate of electrons and holes is effectively suppressed [51,52]. Among all composites, the sample of 0.5 CISI shows the highest TPR intensity, which means an excellent separation efficiency of photogenerated charge carriers for potentially achieving the highest photocatalytic activity.

The linear sweep voltammetry (LSV) plots of the samples are presented in Figure 7c in order to investigate the hydrogen evolution kinetics over various samples. Among all samples, 0.5 CISI possesses the lowest onset potential and the highest cathodic current density, which is advantageous for electrocatalytic hydrogen evolution reaction. Thus, the formation of the heterojunction is beneficial to the decreased onset potential and enhanced kinetics for hydrogen evolution, indirectly manifesting the increased reduction capacity.

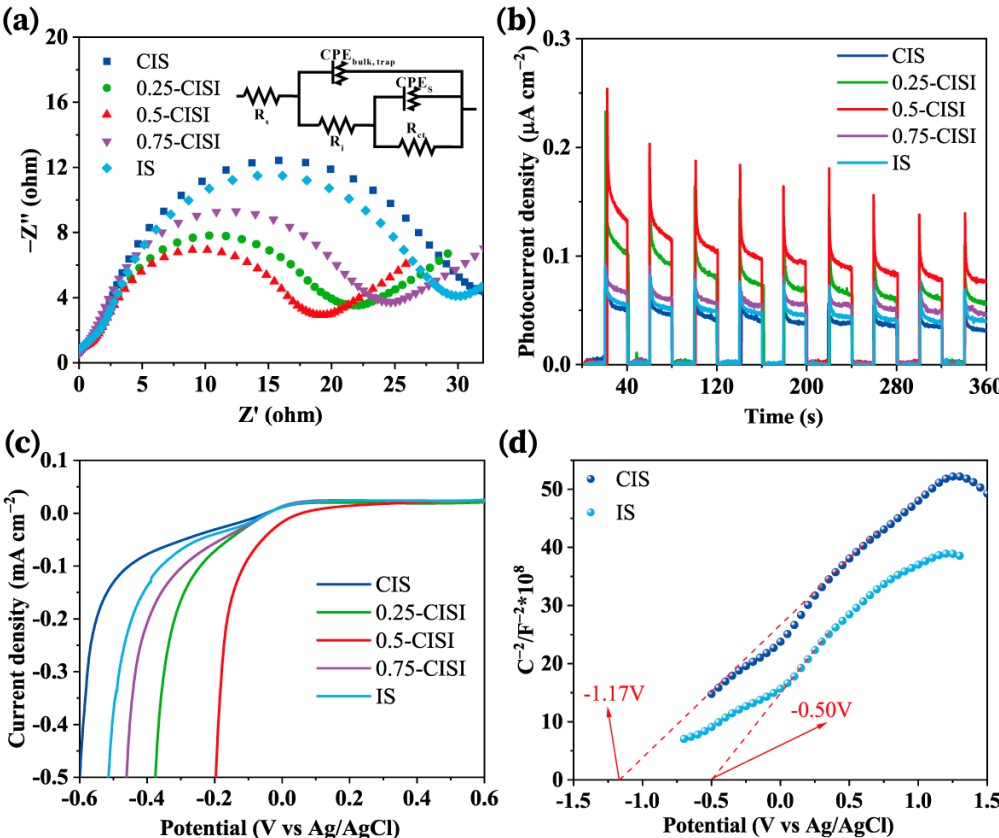

**Figure 7.** (**a**) Electrochemical impedance spectra (EIS) (inset is the equivalent circuit to fit the plots, where $R_s$ represents the overall series resistance of the circuit, $R_i$ represents the internal resistance of the electrode along with a constant phase element ($CPE_{bulk, trap}$), and $R_{ct}$ denotes the charge transfer resistance from the surface states to the solution with the corresponding $CPE_s$). (**b**) Transient photocurrent responses (TPRs). (**c**) Linear sweep voltammetry (LSV) plots. (**d**) Mott–Schottky curves of as-prepared samples measured at a frequency of 1000 Hz.

The matched band structure of photocatalysts determines their Cr(VI) photoreduction performance. Mott–Schottky (M-S) plots of CIS and IS are measured at a frequency of 1000 Hz to determine their band positions, as displayed in Figure 7d. Based on the X-axis tangents intercept, the flat-band potentials ($E_{fb}$) of CIS and IS can be estimated to be –1.17 and –0.50 V (vs. Ag/AgCl at pH = 7.0), respectively. The positive slopes over CIS and IS are their n-type semiconductors, where the conduction band potential ($E_{CB}$) is generally 0~0.2 V more negative than its $E_{fb}$ [53]. Considering the voltage discrepancy of 0.1 V between $E_{CB}$ and $E_{fb}$, the $E_{CB}$ of CIS and IS are calculated as $-1.27$ V and $-0.60$ V (vs. Ag/AgCl at pH = 7.0), respectively [54]. Based on the formula, $E_{NHE} = E_{Ag/AgCl} + 0.059 * pH - E°_{Ag/AgCl}$ ($E°_{Ag/AgCl} = 0.197$ V, pH = 7.0), the $E_{CB}$ of CIS and IS are equal to $-1.05$ eV and $-0.38$ eV versus the normal hydrogen electrode (NHE), respectively. Considering the bandgap values of 2.47 eV for CIS and 2.23 eV for IS, the valence band (VB) potentials ($E_{VB}$) of CIS and IS are estimated to be $+1.42$ eV and $+1.85$ eV (vs. NHE), respectively.

The photocatalytic activity of the as-prepared samples was evaluated by Cr(VI) reduction under visible light irradiation. As shown in Figure 8a, the samples of CIS and IS show relatively low photocatalytic activity due to their rapid recombination rate of the photogenerated charge carrier. After coupling CIS with IS, the CISI composites exhibit a higher Cr(VI) photoreduction activity than CIS and IS. Then, the enhanced photo-activity of the CISI composites is assigned to the efficient charge carrier separation derived from the heterojunction formation between CIS and IS. Correspondingly, the reaction rate constants of all samples are measured with the following order: CIS (0.03554 min$^{-1}$) < IS (0.05578 min$^{-1}$) < 0.75 CISI (0.09358 min$^{-1}$) < 0.25 CISI (0.13313 min$^{-1}$) < 0.5 CISI (0.15561 min$^{-1}$) (Figure 8b,c). No-

tably, the optimal sample of 0.5 CISI has the highest Cr(VI) photoreduction efficiency, indicating that the proper mass ratio between CIS and IS is necessary to achieve the effective heterojunction for boosting photocatalytic activity. To monitor the photocatalytic process of Cr(VI) reduction, the UV-vis spectral changes over 0.5 CISI are displayed in Figure 8d toward the Cr(VI) solution with an absorption characteristic peak at approximately 540 nm. As the irradiation time increases, the peak intensity is gradually decreased and Cr(VI) is almost completely removed within 45 min. In addition, the overall Cr(VI) photoreduction efficiency over 0.5 CISI is higher than that of most reported photocatalysts, as listed in Table 1.

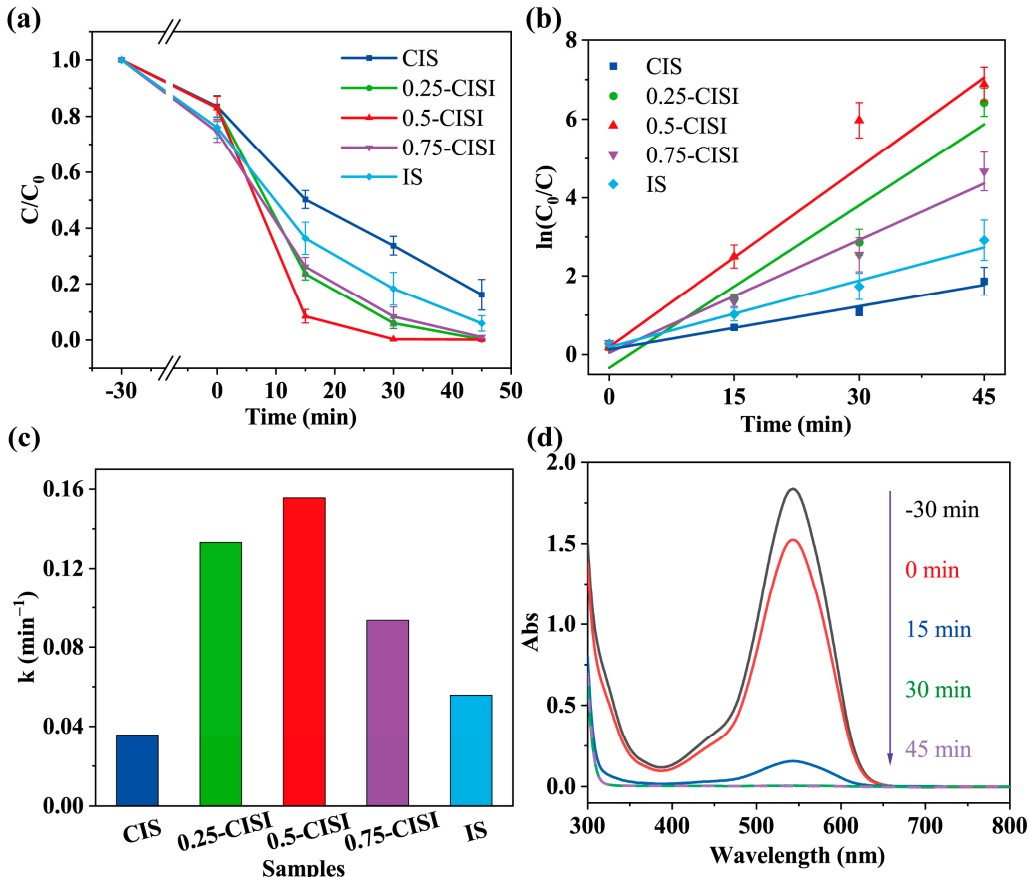

**Figure 8.** (**a**) Photocatalytic Cr(VI) reduction curves, (**b**) the kinetic curves, and (**c**) the kinetic value of samples. (**d**) UV-vis spectral changes for Cr(VI) photo-reduction over 0.5 CISI.

**Table 1.** Comparison of photocatalytic Cr(VI) reduction over different catalysts.

| Catalyst | Catalyst (mg) | Cr(VI) (mg/L) | Solution (mL) | Time (min) | Refs. |
|---|---|---|---|---|---|
| Cu(X)–CdIn$_2$S$_4$ | 20 | 20 | 40 | 110 | [55] |
| CdIn$_2$S$_4$/In(OH)$_3$/NiCr-LDH | 30 | 50 | 50 | 120 | [56] |
| CdIn$_2$S$_4$@UiO-66-NH$_2$ | 50 | 20 | 100 | 120 | [57] |
| Bi$_2$S$_3$-In$_2$S$_3$ | 20 | 70 | 100 | 140 | [58] |
| In$_2$S$_3$/TiO$_2$ | 50 | 20 | 100 | 80 | [59] |
| Mo$_2$C/MoS$_2$/In$_2$S$_3$ | 50 | 40 | 50 | 90 | [60] |
| In$_2$S$_3$/G | 15 | 10 | 30 | 50 | [61] |
| In$_2$S$_3$/BiOBr | 200 | 50 | 500 | 180 | [62] |
| This work | 10 | 20 | 100 | 45 | - |

The cycling experiments for the Cr(VI) photoreduction were conducted to evaluate the photocatalytic stability of the catalyst. As shown in Figure 9a, after five cycles under the same condition, the sample of 0.5 CISI still exhibits the highly photocatalytic efficiency. As

shown in Figure 9b, after Cr(VI) photoreduction cycling experiments, the XRD pattern of 0.5 CISI shows no obvious change but does show the slightly decreased intensity compared to the fresh 0.5 CISI sample, indicating the highly structural stability. Additionally, SEM images of 0.5 CISI before (Figure 1d) and after (Figure 9c) Cr(VI) photoreduction are similar to each other. These results verify the excellent structural stability of 0.5 CISI, which demonstrates good reusability in practical application.

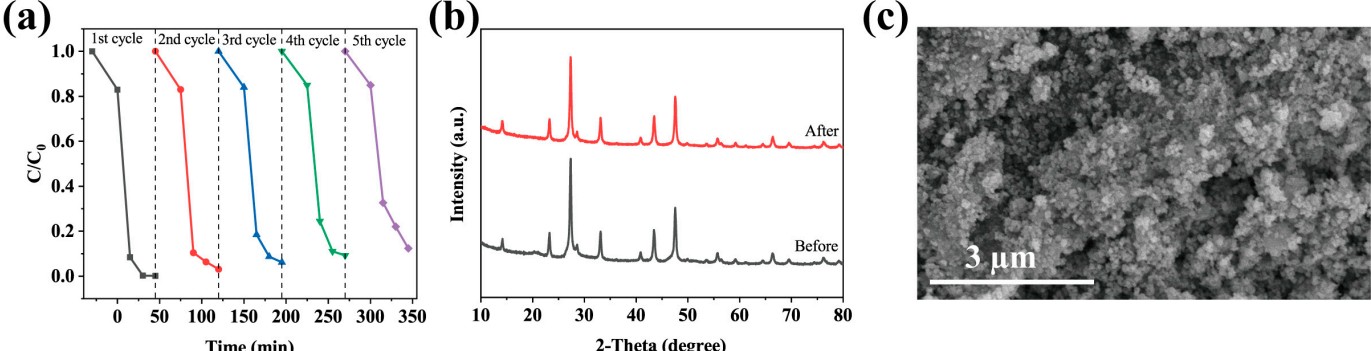

**Figure 9.** (**a**) Cyclic performance of 0.5 CISI for the reduction of aqueous Cr(VI) under visible light irradiation. (**b**) XRD patterns of 0.5 CISI before and after Cr(VI) photoreduction for five cycles. (**c**) SEM image of 0.5 CISI after Cr(VI) photoreduction for five cycles.

In light of the above results and discussion, a possible mechanism for Cr(VI) photoreduction was proposed for 0.5 CISI (Figure 10). A one-step synthesis approach is beneficial to the formation of an intimate interface between CIS and IS in the 0.5 CISI composite. Under visible light, the components of CIS and IS in 0.5 CISI are excited to produce electron and hole pairs. Considering the potential differences, the photogenerated electrons are quickly migrated from CB of CIS to CB of IS across the intimate coupling interface. Meanwhile, an opposite pathway for the transport of photogenerated holes appears from VB of IS to VB of CIS. Subsequently, parts of the powerful electrons in CB of IS contribute to the formation of $^{\bullet}O_2^{-}$. Consequently, these generated $^{\bullet}O_2^{-}$ and $e^{-}$ radicals mainly participate in Cr(VI) photoreduction.

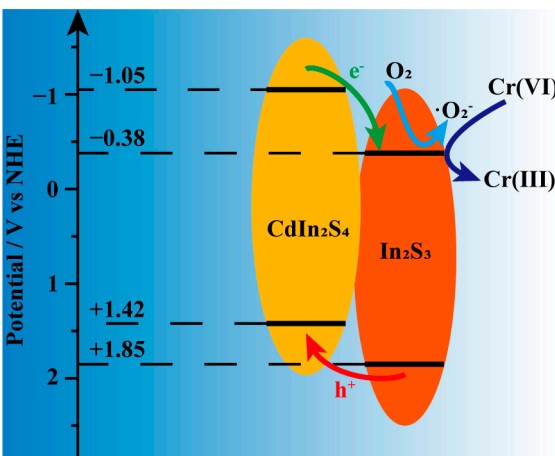

**Figure 10.** Photocatalytic mechanism for Cr(VI) reduction on 0.5 CISI.

## 4. Conclusions

In this work, $CdIn_2S_4/In_2S_3$ (CISI) heterostructured photocatalysts were constructed by a one-step refluxing process, showing an intimate interface between CIS and IS. This formed tight interface between two components could effectively promote the separation and migration of photogenerated electrons and holes. Compared to CIS and IS, the resulting

CISI composites exhibited the enhanced activity for visible-light-driven Cr(VI) photoreduction due to the heterojunction formation between CIS and IS. The optimal sample of 0.5 CISI exhibited the highest Cr(VI) photoreduction efficiency and a reaction rate constant of 0.15561 min$^{-1}$ for the complete removal Cr(VI) within 45 min. This implies that the proper mass ratio between CIS and IS could achieve the effective heterojunction for boosting photocatalytic activity. The results of the PL, TR-PL, EIS, and TPR curves significantly verified that the heterojunction formation between CIS and IS could effectively promote the separation and migration of photogenerated electrons and holes, leading to highly photocatalytic activity. Based on the characterization results, a possible photocatalytic mechanism was speculated for Cr(VI) reduction.

**Author Contributions:** Conceptualization, J.H. and C.L.; methodology, J.W.; software, J.H.; validation, X.L., C.L. and Q.Z.; formal analysis, J.H.; investigation, S.Z.; resources, C.L.; data curation, J.H.; writing—original draft preparation, W.X.; writing—review and editing, C.L.; visualization, W.C.; supervision, X.L.; project administration, H.H.; funding acquisition, C.L. All authors have read and agreed to the published version of the manuscript.

**Funding:** This research was funded by the National Natural Science Foundation of China, grant numbers No. 51902282 and 12274361. This work was also supported by the Qinglan Project of Jiangsu of China, the Natural Science Foundation of Jiangsu Province (BK20211361), and the College Natural Science Research Project of Jiangsu Province (20KJA430004).

**Data Availability Statement:** Data are available on request from the authors.

**Conflicts of Interest:** The authors declare no conflict of interest.

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
