# Peer review of "One-Pot Fabrication of 2D/2D CdIn2S4/In2S3 Heterojunction for Boosting Photocatalytic Cr(VI) Reduction"

_catalysts, doi:10.3390/catal13050826_

Round 1

Reviewer 1 Report

The resulted CISI composites exhibited the highly visible-light-responsive photocatalytic activity for Cr(VI) reduction, with the optimal sample of 0.5-CISI. Please comparison with others in detail 

Reviewer 2 Report

The manuscript deals with the One-pot fabrication of 2D/2D CdIn2S4/In2S3 heterojunction for 2 boosting photocatalytic Cr(VI) reduction. But some major modifications are necessary before its acceptance.

1. There are many grammatical mistakes, please check the manuscript for grammar and English.

2. What is the novelty of the present work? Rewrite it at the end of the introduction section.

3. Add the error bar in Fig. 8

4. Compare your photocatalytic results with other researchers' work in tabular form.

5. Please add the stability of the photocatalytic experiment.

6.The samples after cyclic reactions need to be compared with fresh samples, such as SEM and XRD.

7. In this study, the surface area is a significant parameter. Therefore, I strongly suggest adding the BET surface area measurement for all samples in the revised manuscript or supporting information.

8. To enrich literature add some literature i. Journal of Hazardous Materials 419, (2021), 126453, ii. Recent Patents on Nanotechnology 17 (1), (2023), 5-7, iii. Journal of Alloys and Compounds 928, (2022), 167133.

Reviewer 3 Report

The paper is devoted to the fabrication of 2D/2D CdIn2S4/In2S3 heterojunction for boosting photocatalytic Cr(VI) reduction.  The concept is scientifically sound and quite interesting.

However, there are some defects which need to be corrected. Thereby, I would recommend the manuscript for publication in Catalysts after revision. Some comments are provided as follows:

1.Figure 2 g-j – Please add the scale bar to the images.

2. Figure 5b – In Cd3d XPS spectrum of 0.5-CISI the signal of Cd was not detected. The reason of this should be explained.

3.Figure 6b – The average lifetimes of other CISI composites (0.25-CISI and 0.75-CISI) should be calculated and compared.

4. Figure 7a - Equivalent circuit modelling of the electrochemical impedance spectra should be performed among with resistance values calculation.  

5. Figure 7b - Authors claim that the increased TPR intensity of 0.5-CISI composite suggests that the recombination rate of electrons and holes is effectively suppressed. However, the high positive photocurrent spikes are observed indicating that the high surface charge recombination still exists. Please assess quantitatively the carrier lifetime from the transient photocurrent measurements for better understanding the highest performance of 0.5-CISI composite.

6. Authors conclude that the formation of heterojunction is beneficial to the decreased onset potential and enhanced kinetics for hydrogen evolution, indirectly manifesting the increased reduction capacity. It is better to suggest the reason of the enhanced activity of heterojunction in HER and the large difference in the onset potential for 0.25-CISI, 0.5-CISI and 0.75-CISI.

Moreover, in the LSV curves  (Figure 7c) the weak anodic peak is observed in CIS, IS, 0.25-CISI, 0.75-CISI samples. What is the reason of this peak? However, there is no peak in the LSV curve for 0.5 CISI. Why?

7. Page 3 (2.6. Photocatalytic Cr(VI) reduction) – What solution was used as a source of Cr(VI)?

Reviewer 4 Report

Dear Authors,

I have reviewed your Review manuscript “One-pot fabrication of 2D/2D CdIn2S4/In2S3 heterojunction for boosting photocatalytic Cr(VI) reduction” and expressed my positive feedback regarding your submission. You have explored a very interesting and important topic, but it should be pointed out more, so I request major revisions.

My comments are as follows:

· Figure 8. The green line (30 min of degradation) is below purpura (45 min of degradation), and it seems the authors forgot to draw the green line. In my opinion, the purpura line should be chopped (like this - - - - - - - - -), and then the green line will be visible.

· The morphology should be provided after the cyclic test or at least after the degradation test. A reusability test should be performed.

· Detailed information on the standard deviation error bars in data collection in photocatalytic application studies should be provided.

Best regards

Round 2

Reviewer 2 Report

The revision made by the author is satisfactory. The present form of the manuscript should be accepted.

Reviewer 3 Report

The paper can be accepted for publication in the revised form.

Reviewer 4 Report

Dear Authors,

You have completed all required corrections, and I believe the paper is suitable for publication.